# An Observational Study Comparing Fibromyalgia and Chronic Low Back Pain in Somatosensory Sensitivity, Motor Function and Balance

**DOI:** 10.3390/healthcare9111533

**Published:** 2021-11-10

**Authors:** José Antonio Mingorance, Pedro Montoya, José García Vivas Miranda, Inmaculada Riquelme

**Affiliations:** 1Research Institute of Health Sciences (IUNICS-IdISBa), University of the Balearic Islands, 07122 Palma de Mallorca, Spain; inma.riquelme@uib.es; 2Physiotherapy Department, Son Espases Hospital, 07120 Palma de Mallorca, Spain; 3Department of Nursing and Physiotherapy, University of the Balearic Islands, 07122 Palma de Mallorca, Spain; 4Center for Mathematics, Computing and Cognition, Federal University of ABC, São Bernardo do Campo 09606-070, Brazil; 5Laboratory of Biosystems, Institute of Physics, Federal University of Bahia, Salvador 40170-115, Brazil; vivasm@gmail.com

**Keywords:** fibromyalgia, low back pain, sensitivity, proprioception, balance, gait

## Abstract

Fibromyalgia (FM) and chronic low back pain (CLBP) have shared pathophysiology and have a considerable impact on patients’ daily activities and quality of life. The main objective of this study was to compare pain impact, somatosensory sensitivity, motor functionality, and balance among 60 patients with FM, 60 patients with CLBP, and 60 pain-free controls aged between 30 and 65 years. It is essential to know the possible differences existing in symptomatology of two of the major chronic pain processes that most affect the population, such as FM and CLBP. The fact of establishing possible differences in sensory thresholds, motor function, and proprioceptive measures among patients with FM and CLBP could bring us closer to a greater knowledge of the chronic pain process. Through an observational study, a comparison was made between the three groups (FM, CLBP, and pain-free controls) evaluating functional performance, postural balance, kinematic gait parameters, strength, depression, fatigue, and sensitivity to pain and vibration. Patients with chronic pain showed worse somatosensory sensitivity (*p* < 0.001) and motor function (*p* < 0.001) than pain-free controls. Moreover, patients with FM showed greater pain impact (*p* < 0.001) and bigger somatosensory (*p* < 0.001) and motor deficiencies (*p* < 0.001) than patients with CLBP. Further research should explore the possible reasons for the greater deterioration in patients with FM in comparison with other chronic pain conditions. Our results, showing the multiple areas susceptible of deterioration, make it necessary to adopt interdisciplinary interventions focused both on physical and emotional dysfunction.

## 1. Introduction

Chronic pain represents one of the most important public health problems, accounting for significant personal, social, and economic burdens. Fibromyalgia (FM) and chronic low back pain (CLBP) are chronic pain conditions with high frequent medical consultation in primary care and a high negative impact on function and quality of life [1].

FM is a chronic disease of idiopathic etiology, which is estimated to affect 2–4% of the population [2]. It causes pain, stiffness, and tenderness in muscles, tendons, and joints, as well as sleep disturbances, fatigue, anxiety, depression, and alterations in intestinal functions [2]. CLBP is a pain of more than 12 weeks limited to the lower area of the back. Musculoskeletal dysfunctions, as in the intervertebral disc, zygapophysial joint, and sacroiliac joint, are believed to be common pain generators of CLBP [3]. 

FM and CLBP present common characteristics in comparison with pain-free controls. Thus, both conditions show hyperalgesia, signs of central pain sensitization, increased excitability in the central nervous system, and abnormal endogenous pain modulation [1]. Moreover, both pathologies have been associated with deficits in motor function and motor control mechanisms, since long-lasting nociceptive interferences can cause long-term neurological adaptations of postural and motor behavior [4,5]. In this sense, both pathologies have been associated with dynamic balance impairments (anticipatory postural adjustments and steady-state balance), which are related to changes in body perception [6,7,8,9] and to high frequency of falls [10,11,12,13,14]. Furthermore, it has been observed that both FM and CLBP patients have similar gait abnormalities, such as biomechanical disturbances and higher metabolic demands and fatigue in comparison with healthy controls [11,15,16,17,18,19].

Despite the clinical similarities between FM and CLBP, little is known about the differences between these two chronic pain conditions in terms of somatosensory and motor symptoms. Thus, for example, it is unknown whether gait or balance are more impaired in FM than in CLBP or vice versa. To our knowledge, there are no studies that directly compare FM and CLBP patients on motor function, balance, and somatosensory parameters. Thus, the major aim of the present study was to compare patients with FM and CLBP regarding some clinical symptoms (pain intensity, pain descriptors, depression, anxiety, fatigue, stiffness, and perceived sleep disturbances), as well as regarding parameters of pain and vibration sensitivity, functional gait, and balance. Based on previous evidence indicating that CLBP in many patients preceded the typical generalized pain of FM [1], we hypothesized that the alterations in patients with FM would be greater than in patients with CLBP. For a better understanding of the disturbances in both groups of chronic pain patients, a group of pain-free participants was also analyzed using the same study protocol.

## 2. Materials and Methods

### 2.1. Ethics

This observational study was approved by the Research Ethics Committee of the Balearic Islands (Spain) (Reference: IB-2586/15 PI) and was conducted in accordance with the Declaration of Helsinki of 1975. All subjects included in the study signed the informed consent form.

### 2.2. Participants

Patients with FM and patients with CLBP, as well as pain-free controls matched in age, gender, and sociodemographic characteristics, were recruited from different health centers and patients’ associations in Majorca (Spain), through publicity talks, during the months of January and February 2016. Sixty patients with FM, 60 patients with CLBP, and 60 pain-free controls agreed to participate in the study. Inclusion criteria were: (1) age between 30 and 65 years and (2) clinical diagnosis of FM or CLBP, or pain-free healthy volunteers. Patients with FM were included in the study if they fulfilled the criteria of the American College of Rheumatology for fibromyalgia [18]. Only low-back-pain patients in whom axial back pain was the predominant complaint were included in the study. Regarding the clinical characteristics of the participants with chronic pain, patients were included if (a) their symptoms lasted at least six months, with a minimum intensity of 50/100 mm on a visual analog pain scale; (b) they had no history of previous spinal surgery [3]; and (c) their physical disability was less than 40%. Pain-free volunteers were included if they did not present pain symptoms or some type of treatment in any part of the body during the previous 12 months. Participants were excluded from the study if they had not signed the informed consent or if they reported any other musculoskeletal disorder rather than FM or CLBP, any neurological disorder, or had previous spinal fusion surgery or spinal cord stimulation. Patients with pain radiating into the leg or any other low limb location were excluded to ensure a homogenous group.

### 2.3. Instruments and Procedure

The assessment session was performed in a spacious and silent room (14 m × 12 m) in the Fibromyalgia Association of Palma (Spain) by two investigators, members of the research team. After informed consent was signed, participants firstly completed the self-report questionnaires and secondly performed the motor function and sensitivity tasks. Finally, the static and dynamic balance task was video recorded. Four types of measures were obtained: self-report questionnaires, motor function, somatosensory sensitivity, and static and dynamic balance. Evaluations were carried out in the same order for all participants. The complete assessment lasted about one hour and was performed during the months of March, April, May, and June 2016.

### 2.4. Self-Report Questionnaires

*Fibromyalgia Impact Questionnaire (FIQ)* [20]. This is an extensively validated instrument designed to quantify the overall impact of fibromyalgia over many dimensions (e.g., pain level, fatigue, anxiety, depression, etc.). A higher score indicates a greater impact on the person. The total FIQ score is between 0 and 100, with 0 representing the highest functional capacity and quality of life and 100 the worst state. All participants completed the questionnaire.

*McGill pain Questionnaire (SF-MPQ)* [21]. This abbreviated form has three subscales (A, B, and C). Subscale A includes items that assess sensory and affective function. Subscale B consists of a visual analog scale, and Subscale C is a verbal descriptor inventory that allows the patient to assign a value for the current pain experience. All participants completed the questionnaire.

### 2.5. Somatosensory Sensitivity

*Pressure pain thresholds* (expressed in newtons) were assessed by means of a standard digital force gauge (Force One, Wagner Instruments) using a flat rubber tip [22]. Pressure stimuli were applied on three bilateral body locations: the great trochanters (related to the pain area in patients with CLBP), epicondyles (related to the pain area in patients with FM), and index finger (not related to pain areas). The mean of three trials per location, without breaks, was calculated, and an average of the right and left sides was computed to have a final score for each location.

*Vibration thresholds* were evaluated by using a Vibratron (Physitemp Instruments, Clifton, NJ, USA) [23]. The Vibratron consists of a controller and two identical transducers that were used to determine the intensity of the vibratory stimulus perceived by the participant. The testing started with vibration intensity above the threshold (easily detected by the patient), and then it was gradually reduced, asking participants to indicate when the vibration was not perceived. Vibration values displayed on the control unit are mean (standard deviation) vibration units, corresponding to the amplitude of vibration (proportional to the square of applied voltage) [24]. The mean of three trials per location, without breaks, was calculated, and an average of the right and left sides was computed to have a final score for each location.

### 2.6. Motor Function

*The Berg scale* [25] is a functional balance assessment tool, consisting of 14 functional tasks scored with values ranging from 0 (cannot perform) to 4 (normal performance). The general scores range from 0 (severely impaired balance) to 56 (excellent balance). This scale has been used in previous studies to assess balance in patients with FM [26] and CLBP [27].

*The six-minute walking test* [28] is a functional test scoring the total distance walked by a person in 6 min. The set-up was a 30-m corridor before the entrance to the room with place cones at either end of the 30-m stretch as turning points. This test has been validated in several populations, including patients with FM [29] and CLBP [30].

*Timed up and go test* [31]. Patients were instructed to sit on a chair with back support, stand up from the chair, walk to a mark located 3 m away, return to the chair, and sit down again. The task was performed only once and measured in seconds, correlating with gait speed, balance, functional level, and ability to go out. This test has been validated in several populations, including patients with FM [32] and CLBP [33].

*Isometric back muscle strength* was determined by using a back muscle dynamometer (Takei Physical Fitness Test T.K.K.5002). Participants were asked to place their feet on the top of one platform, bend forward with 30° of lumbar flexion, and pull by extending their back to put the body as vertical as possible. The average force from two trials was computed [34]. Strength data were presented normalized for body mass [35].

*The Borg scale* [36] is a self-report measure of fatigue and subjective perception of dyspnea. It consists of a 10-point scale ranging from 0 (complete lack of dyspnea or fatigue) to 10 (maximum dyspnea or fatigue). In the present study, ratings were obtained before and after the 6 min walking test.

### 2.7. Static and Dynamic Balance

Static balance was assessed by using a modified version of Romberg’s balance test [37]. This test is based on the fact that balance arises from the combination of several neurological systems (proprioception, vestibular input, and vision) and that maintaining balance while standing in the stationary position with closed eyes should rely on intact sensorimotor integration centers and motor pathways. Thus, the essential feature of the test is that the participant should become unsteadied with eyes closed. In the present study, we analyzed the oscillatory body movements during the test performance. Participants were situated below a webcam situated above the ground and regulated to be at a mean distance of 50 cm from the participant’s head. Participants were asked to remain in an orthostatic position with feet parallel at shoulder height, arms extended along the body, and eyes closed for 1 min [38]. The participant carried a headband on the parietal level with two points separated 5 cm to further analyze the motion parameters (velocity and body sway). Motion on the frontal and sagittal planes was captured with a digital video camera at 210 frames per second. For analyses of motion parameters, a free, open-source software for computer vision analysis of human movement was used (CvMob, 2011) [39]. This software has a high degree of accuracy for calculating body position and movement in the X and Y axes recorded by conventional cameras [40].

*Dynamic balance* was tested by means of a gait task [15,16,41]. Participants were instructed to walk on a 4 m carpet at their normal walking step, with socks and with flexed arms positioned on the abdomen. Optical markers were attached at the following three body locations: area between the lateral condyle of the femur and the fibular head, great trochanter, and lateral malleolus. The subject’s motion was digitally recorded with a video camera at 210 frames per second (CasioExilimEX-FS10). The camera was positioned at a distance of four meters from the carpet to visualize changes in position, velocity, and anatomical points along the x-axis. The CvMob 3.1 software was used to extract the following variables: gait velocity (cm/s), stride length (cm), percentage of time in the stance/swing phase, and percentage of time with single and double support.

### 2.8. Statistical Analyses

Statistical analyses were performed by using the SPSS software. Kolmogorov–Smirnov tests were previously carried out to test the normality of the dependent variables. The null hypothesis that data were sampled from a normally distributed population was examined by using Shapiro–Wilk tests, and differences between patients and pain-free controls were analyzed by using ANOVAs with Group as an intersubject factor and post-hoc comparison analysis by pairs with Bonferroni adjustment. A value of *p* = 0.05 was used for statistical significance.

## 3. Results

Sample size calculation was performed by using the GRANMO sample size calculator (GRANMO: http://www.imim.es/ (accessed on 8/1/2016)). The prevalence of FM in Spain is 2.4% (95% CI: 1.5–3.2), with a 21:1 female/male ratio [19]. The estimated prevalence of people with chronic low back pain among Spanish adults is 7.7% [42]. A sample size calculation was performed taking into account a population of 20,000 patients with FM patients and 30,000 affected by CLBP in the Balearic Islands, with a confidence interval of 95% and an accuracy of the estimate of 5%. In this way, we determined that a total of 60 subjects were required to achieve statistical significance. Participants reported their age, sex, height, weight, and pain duration. Table 1 displays the characteristics of the three groups of participants. Regarding medication intake, most chronic pain participants (patients with FM or CLBP) were taking pain medication such as analgesics, anxiolytics, and antidepressants. For medical and ethical reasons, medication was not discontinued during the study.

The Kolmogorov–Smirnov and Shapiro–Wilk tests indicated that all dependent variables fulfilled the normality assumption for parametric analyses (all *p* > 0.05). In addition, groups were similar in their sociodemographic characteristics (all *p* > 0.05, Table 1). Significant differences in pain duration (F (2,177) = 128.54, *p* < 0.001) confirmed that individuals with chronic pain had longer pain duration than pain-free controls. Post-hoc comparisons showed significant differences between each of the two groups of chronic pain (FM and CLBP) and pain-free controls (both *p* < 0.001), although not between patients with FM and patients with CLBP (*p* > 0.05).

Table 2 displays mean values and *p*-values for the statistical comparisons among the three groups of participants on all dependent variables of the study.

### 3.1. Self-Report Questionnaires

Significant group differences were found in the total scores of the *Fibromyalgia Impact Questionnaire* (F (2,177) = 580.96 *p* < 0.001). Post-hoc comparisons showed significant differences between each of the two groups of chronic pain (FM and CLBP) and pain-free controls (both *p* < 0.001) and objectified higher values in patients with FM than patients with CLBP (*p* < 0.001), indicating a high impact of chronic pain on daily life. We also found significant group differences in the following subscales: pain (F (2,177) = 382.45, *p* < 0.001), fatigue (F (2,177) = 366.65, *p* < 0.001), anxiety (F (2,177) = 486.57, *p* < 0.001), and depression (F (2,177) = 453.24, *p* < 0.001), with post-hoc comparisons showing the same trend (higher scores in both groups of chronic pain compared to pain-free controls and higher scores in patients with FM compared to patients with CLBP). Group differences were also found in the subscales ability to perform tasks (F (2,177) = 486.57, *p* < 0.001), missed job (F (2,177) = 142.83, *p* < 0.001), do job (F (2,177) = 74.7, *p* < 0.001), and stiffness (F (2,177) = 382.29, *p* < 0.001). Post-hoc comparisons showed significant differences between the two chronic pain entities and pain-free controls (both *p* < 0.001) but no significant differences between patients with FM and patients with CLBP (all *p* > 0.05).

Regarding the *McGill Pain Questionnaire*, significant group differences were observed in the three subscales: in the sensory/affective components of Subscale A (F (2,177) = 165.62, *p* < 0.001), in pain intensity measured by a visual analog pain scale (Subscale B) (F (2,177) = 1162, *p* < 0.001), and in the verbal descriptor inventory (Subscale C) (F (2,177) = 1152.7, *p* < 0.001). Post-hoc comparisons also showed significant differences between each of the two groups of chronic pain and pain-free controls (all *p* < 0.001) and that patients with FM reported higher scores than patients with CLBP in all subscales (all *p* < 0.001). Again, higher scores in these subscales indicated greater pain impact in patients with chronic pain than in pain-free controls and higher pain impact in patients with FM than in patients with CLBP.

### 3.2. Somatosensory Sensitivity

For *pressure pain thresholds*, significant effects due to group (F (2,177) = 202.17, *p* < 0.001) and group x body location (F (4,354) = 26.99, *p* < 0.001), but not due to body location (F (2,354) = 2.31, *p* = 0.104), were observed. Post-hoc comparisons revealed that pressure pain thresholds were significantly different at the three body locations within each of the groups (all *p* < 0.05), except between epicondyle and trochanter in fibromyalgia patients (*p* = 0.802). Post-hoc comparisons also showed significant differences between the two groups of chronic pain and healthy people in epicondyles, greater trochanters, and index fingers (all *p* < 0.001); these analyses also showed that patients with FM were more sensitive at epicondyles and greater trochanters than patients with CLBP (all *p* < 0.001). However, no significant differences between patients with FM and patients with CLBP were found in index fingers (*p* > 0.05) (Figure 1).

For *vibration thresholds,* significant main effects of body location (index finger vs. toes) were observed (F (1,177) = 703.64, *p* < 0.001), with higher thresholds for toes. Moreover, a significant interaction effect of group x body location was observed (F (2,177) = 17.21, *p* < 0.001). Post-hoc comparisons revealed that pain-free controls were more sensitive (lower thresholds) to vibration stimuli in the two locations than both groups with chronic pain (all *p* < 0.001). Moreover, it was found that patients with FM were less sensitive than patients with CLBP both at the toes (*p* = 0.032) and the index finger (*p* = 0.042) (Figure 2).

### 3.3. Motor Function

In general, patients with chronic pain performed worse than pain-free controls and patients with FM performed worse than patients with CLBP in all measures of motor function (*Berg scale*, *timed up and go* test, *six-minute walking* test, *Borg scale,* and *isometric back muscle strength*). Thus, significant group differences were found in performance scores of the *Berg scale* (F (2,177) = 458.68, *p* < 0.001), showing an impaired motor function of the two groups of chronic pain compared to pain-free controls (both *p* < 0.001) and worse performance in patients with FM than in patients with CLBP (*p* < 0.005). Statistical analyses of the *six-minute walking* test also revealed significant differences among groups (F (2,177) = 183.82, *p* < 0.001), with both chronic pain groups walking less distance than pain-free controls (all *p* < 0.001) and patients with FM walking less distance than patients with CLBP (*p* < 0.005). In a similar way, significant group differences in the performance of the *timed up and go* test (F (2,177) = 122.34, *p* < 0.001) indicated the faster performance of the pain-free controls compared to the chronic pain groups (both *p* < 0.005), and faster performance in the CLBP than in the FM group (*p* < 0.001). Ratings on self-perceived effort obtained from the *Borg scale* after the *six-minute walking* test revealed significant group differences in fatigue (F (2,177) = 225.9, *p* < 0.001); post-hoc analysis showed that fatigue was higher in the groups of chronic pain than in healthy controls (both *p* < 0.001) and higher in patients with FM than in patients with CLBP (all *p* < 0.001). Finally, significant group differences in *isometric back muscle strength* (F (2,177) = 296.80, *p* < 0.001), showed lower strength in patients with FM and patients with CLBP compared with pain-free controls (all *p* < 0.001) and in patients with FM compared with patients with CLBP (*p* < 0.05).

### 3.4. Static and Dynamic Balance

*Static balance*. Three parameters were obtained from the analyses of Romberg’s static balance task: mean velocity and standard deviations of anteroposterior and mediolateral sway. Significant group differences were observed in mean velocity (F (2,177) = 50.52, *p* < 0.001) and standard deviations in anteroposterior (F (2,177) = 23.4, *p* < 0.001) and mediolateral sway (F (2,177) = 31.18, *p* < 0.001). Post-hoc analysis showed that both groups of chronic pain displayed higher scores of mean velocity in comparison with pain-free controls (both *p* < 0.001), and patients with FM displayed greater scores of mean velocity than patients with CLBP (*p* < 0.001). Moreover, both groups of chronic pain displayed greater body sway than controls in the anteroposterior (both *p* < 0.05) and mediolateral directions (both *p* < 0.01), and patients with FM displayed greater scores than patients with CLBP in both directions (both *p* < 0.001). Figure 3 displays the mean of the mediolateral body sway (axis X) and mean of the anteroposterior body sway (axis Y) in the three groups.

*Dynamic balance*. Analyses of gait task kinematic parameters indicated that patients with FM and patients with CLBP had significant deficits in dynamic balance and gait performance compared to pain-free controls. Significant group differences were observed in gait velocity (F (2,177) = 13.25, *p* < 0.001), stride length (F (2,177) = 11.54, *p* < 0.001), single support percentage (F (2,177) = 8.18, *p* < 0.001), and percentage of swing phase (F (2,177) =5.81, *p* < 0.001). Post-hoc comparisons revealed that patients with FM and patients with CLBP displayed reductions in these parameters in comparison with pain-free controls (all *p* < 0.02), but no differences were observed between patients with FM and patients with CLBP (all *p* > 0.05).

## 4. Discussion

The aim of the present study was to compare pain impact, somatosensory sensitivity, motor functionality, and static and dynamic balance in two groups of patients with chronic pain (fibromyalgia and chronic low back pain) and pain-free controls. Based on previous evidence that objectified chronic pain impairments in motor functionality, balance, and somatosensory sensitivity [1,8], it was hypothesized that patients with chronic pain would have worse sensitivity and motor function than pain-free people. Furthermore, based on the research considering CLBP as a pre-stage to FM [43], it was also hypothesized that patients with FM would show a greater impact of pain, and greater somatosensory and motor deficiencies than patients with CLBP. Both hypotheses were confirmed by our results.

As expected, both groups of chronic pain scored worse results than healthy pain-free controls in all the sensory and motor variables. Similar to previous studies, our patients with chronic pain reported greater sensitivity to pain [44], together with reduced sensitivity to nonpainful stimuli (vibration) than pain-free controls [45,46]. The evaluation of motor function also showed worse motor performance and poorer static and dynamic balance in chronic pain patients compared to pain-free controls, also in line with previous research [6,17]. These findings may reflect processes of pain sensitization and pain-related plastic changes in M1 and other cortical motor areas [47,48]. Pain-related central disturbances would affect postural control and the planification of synergistic muscle activation and recruitment to maintain joint stability and movement control [49,50]. In this sense, it has been objectified as a direct relation between M1 functional reorganization (changes of M1 maps) and the delay of trunk muscle activation to control for postural perturbation during focal limb movement [51]. Considering that static and dynamic balance is a complex task that involves the rapid and dynamic integration of multiple sensory, motor, and cognitive inputs to execute appropriate neuromuscular activity, and that M1 integrates information from adjacent sensorimotor areas before launching the motor command towards the spinal motoneurons, functional or connectivity alteration in these areas may lead to worse motor performance in patients with chronic pain.

In the present study, CLBP was encompassed between pain-free controls and FM. Thus, although patients with CLBP showed some indicators of altered pain processing, they did not reach the extent of patients with FM. Particularly, patients with CLBP reported less pain impact and lower pain sensitivity than patients with FM, in accordance with other studies reporting higher deep-tissue hypersensitivity, lower pain tolerance, and higher temporal summation of pain stimuli in FM compared to CLBP [1]. Interestingly, in our study pain sensitivity differences appeared in locations related to pain (even in a typical low back pain location such as the great trochanter), but not in a location unrelated to pain such as the index finger. This finding could be related to the generalized hyperalgesia present in these two chronic pain conditions [52].

In addition, this study showed significant differences in motor performance and static and dynamic balance between both groups of patients with chronic pain, highlighting the greater severity of motor and balance alterations in patients with FM. Poor balance has been considered a predictor of widespread musculoskeletal pain [53]. The lack of differences between patients with CLBP and patients with FM in dynamic, but not in static balance, might reflect wider plastic processes of pain chronification in this latter condition. Various reasons could explain the presence of higher symptom severity in patients with FM compared to patients with CLBP. FM is concomitant to altered central processing of pain stimuli without recognizable peripheral nerve dysfunction or sources of nociceptive input, whereas a more located segmental alteration associated with peripheral sensitization could be more relevant in CLBP [54]. In this sense, other studies have reported that opioid neurotransmitter levels in cerebrospinal fluid were inversely correlated to pain thresholds, which could reflect a higher dysfunction of the endogenous pain inhibitory system in FM compared to CLBP [55]. In addition, patients with FM in our study reported higher scores in the subscales depression and anxiety of the Fibromyalgia Impact Questionnaire and higher scores related to affective descriptors in the McGill Pain Questionnaire. Other authors have also suggested a stronger affective pain component, with higher anxiety, psychological load, and dramatic connotations in the narrative of symptoms onset in patients with FM than with CLBP [56,57].

Depression and anxiety are common predictors of widespread pain [43], opioid misuse [58], and poor pain coping [59]. Moreover, psychological measures, such as emotional or psychosocial distress and somatic awareness, are common phenotypic markers of pain amplification [60]. The psychological attributions for somatic symptoms and the difficulty in emotion description are related to increased anxiety in patients with FM in comparison to patients with CLBP [57]. In consequence, patients with FM may interpret stressful situations as more threatening, increasing pain-related catastrophizing and reducing the adoption of positive coping strategies (e.g., problem solving) [59]. Higher FM scores for somatization added to the high prevalence of pain in different locations might lead to a wide-reaching dysregulation of autonomic and hypothalamic–pituitary–adrenal axis function [61,62]. This dysregulation has been characterized in FM by mild hypocortisolemia, hyperactivity of pituitary ACTH release to CRH, and glucocorticoid feedback resistance, while only mild dysregulation signs, as hypercortisolemia, have been found in CLBP [62]. This abnormal stress response in FM could trigger aberrant glial activity and promote additional factors for chronic pain severity [43]. Furthermore, peripheral blood mononuclear cell beta-endorphin concentration is decreased in chronic fatigue syndrome and fibromyalgia, but not in depression [63]. Likewise, it has been observed that depression is associated with greater affective dysregulation and a deficit in information processing speed in fibromyalgia [64]. The motor response is slower in highly depressed patients with fibromyalgia than in pain-free controls [65].

Our findings point to the need to establish a more precise diagnosis in chronic pain processes that include the analysis of specific motor, somatosensory, and balance parameters. Nevertheless, some limitations must be considered for the adequate interpretation of the present results. Although medication was controlled, it was not suppressed in participants with chronic pain, and opioids, tricyclics, hypnotics, or benzodiazepines have been demonstrated to have side effects on postural stability. A further limitation of the study is the heterogeneity of pain conditions, especially low back pain, that may have affected the test scores. Moreover, anxiety and depression were scored as subscales of a specific questionnaire for FM, which could have affected the results, especially when low back pain was assessed. Although most patients were postmenopausal, our study did not consider the possible effects of the phase of the menstrual cycle on pain sensitivity. Finally, the fact that most of the participants were women and that especially in the FM group the rate of women was higher than the CLBP could have biased some of the results. Therefore, future research must address these biases and elucidate whether the greater impact of FM on somatosensory sensitivity, motor function, and balance compared to CLBP could be mediated by some of these factors.

## 5. Conclusions

Patients with chronic pain (such as fibromyalgia and chronic low back pain) showed worse pain impact, somatosensory sensitivity, motor function, and balance than pain-free controls. Furthermore, although FM and CLBP may have similar symptoms and share a common pathophysiology, our findings revealed that FM patients showed a greater pain impact and more somatosensory, motor, and balance disturbances than did CLBP patients. Some authors consider CLBP as a stage prior to FM, pointing to pain severity as the most important clinical risk factor for the transition to FM. Indeed, a high percentage of patients with CLBP develop FM where back pain is no longer dominant. This fact highlights the need for early intervention to mitigate the severity of pain symptoms and to alter the course towards FM. Because FM is a prevalent chronic pain syndrome with few effective therapeutic options available, once the possible reasons for the greater deterioration in patients with FM are known, compared to other chronic pain entities such as CLBP, it is necessary to adopt effective treatments in order to prevent it. Our results show that there are multiple areas susceptible to deterioration, so it is necessary to adopt interdisciplinary interventions focused on both physical and emotional dysfunction. Interventions that integrate somatic, physical, and emotional factors into the same rehabilitation program should be considered when developing clinical programs.

## Figures and Tables

**Figure 1 healthcare-09-01533-f001:**
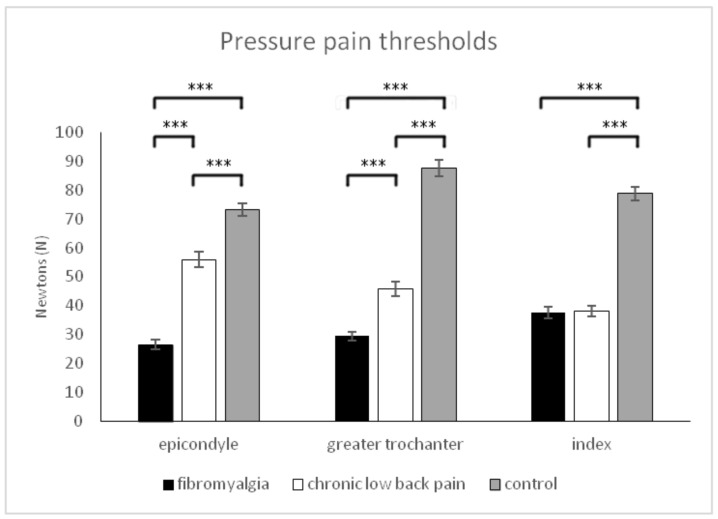
Means and typical errors for pressure pain thresholds at epicondyle, greater trochanter, and index finger in the three groups of participants (fibromyalgia, chronic low back pain and pain-free controls). *** *p* < 0.001.

**Figure 2 healthcare-09-01533-f002:**
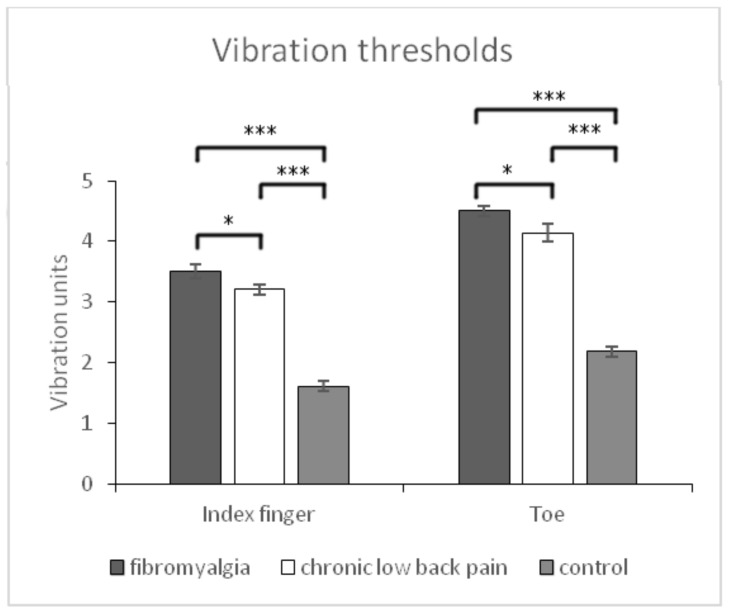
Means and typical errors for vibration thresholds at toe and index finger in the three groups of participants (fibromyalgia, chronic low back pain, and pain-free controls). * *p* < 0.05, *** *p* < 0.001.

**Figure 3 healthcare-09-01533-f003:**
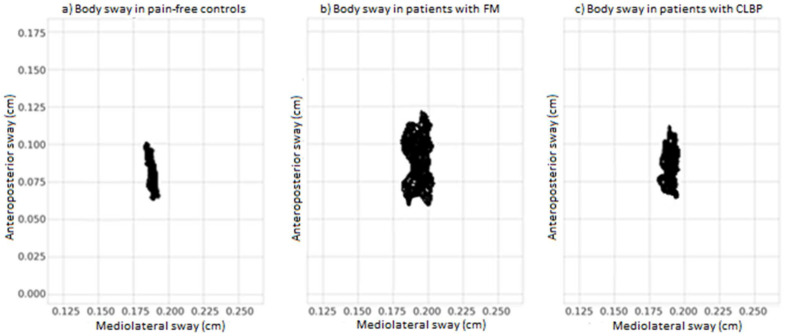
Diagrams of the mean body sway in the anteroposterior and mediolateral directions for the three groups of participants (pain-free controls, fibromyalgia and chronic low back pain). (**a**) Body sway in pain-free controls. (**b**) Body sway in patients with FM. (**c**) Body sway in patients with CLBP.

**Table 1 healthcare-09-01533-t001:** Sociodemographic data from the three groups of participants.

Groups of Participants	FM (*n* = 60)	CLBP (*n* = 60)	Pain-Free Controls (*n* = 60)				
	Mean ± sd	Mean ± sd	Mean ± sd	*p*-values all groups	*p*-values FM—CLBP	*p*-values FM—pain-free controls	*p*-values CLBP—pain-free controls
Age (years)	52.57 ± 1.08	52.50 ± 1.42	49.87 ± 1.25	0.06	0.90	0.05	0.06
Pain duration (years)	7.38 ± 2.79	7.08 ± 4.07	0	<0.001	0.81	<0.001	<0.001
BMI (Kg/m^2^)	23.79 ± 0.34	23.66 ± 0.30	23.15 ± 0.30	0.33	0.77	0.17	0.24
Height (cm)	168.1 ± 0.88	170.36 ± 0.83	168.65 ± 0.76	0.13	0.08	0.64	0.13
Weight (kg)	67.13 ± 0.89	68.65 ± 0.93	65.90 ± 1.00	0.12	0.24	0.33	0.08
Gender	54 ♀ 6 ♂	45 ♀ 15 ♂	45 ♀ 15 ♂	0.06	1	0.06	0.06

FM: fibromyalgia; CLBP: chronic low back pain; BMI: body mass index.

**Table 2 healthcare-09-01533-t002:** Comparisons among the three groups of participants on somatosensory sensitivity, motor function, and balance.

Dependent Variables	FM (*n* = 60)	CLBP (*n* = 60)	Pain-Free Controls (*n* = 60)	*p*-Values					
	Mean ± sd (Range)			All Groups	FM—CLBP	FM—Pain-Free Controls	CLBP—Pain-Free Controls	η	Cohen’s d
Fibromyalgia impact questionnaire (0–100)	80.3 ± 11.49 (77.49–83.11)	59.66 ± 14.22 (56.85–62.47)	13.36 ± 5.46 (10.55–16.16)	*p <* 0.001	*p <* 0.001	*p <* 0.001	*p <* 0.001	0.868	1.597
McGill Pain Questionnaire, Subscale A (0–30)	210.05± 9.88 (19.41–22.68)	12.85 ± 4.96 (11.21–14.28)	0.53 ± 1.18 (–1.10–2.17)	*p <* 0.001	*p <* 0.001	*p <* 0.001	*p <* 0.001	0.637	10.049
McGill Pain Questionnaire, Subscale B (0–10)	7.95± 0.85 (7.63–8.25)	6.26 ± 1.42 (5.96–6.57)	0.49 ± 1.26 (0.18–0.79)	*p <* 0.001	*p <* 0.001	*p <* 0.001	*p <* 0.001	0.877	1.419
McGill Pain Questionnaire, Subscale C (0–5)	40.00± 0.37 (3.54–4.45)	2.75 ± 0.70 (2.29–3.20)	0.97 ± 2.97 (0.45–1.35)	*p <* 0.001	*p <* 0.001	*p <* 0.001	*p <* 0.001	0.342	2.224
Pressure pain sensitivity epicondyles (N) (0–100)	26.43 ± 13.46 (220.02–30.86)	55.95 ± 20.71 (51.53–60.37)	73.26 ± 17.11 (68.84–77.68)	*p <* 0.001	*p <* 0.001	*p <* 0.001	*p <* 0.001	0.558	−1.690
Pressure pain sensitivity greater trochanters (N) (0–100)	29.47 ± 11.59 (24.77–34.17)	45.78 ± 20.35 (410.08–50.48)	87.59 ± 21.74 (82.89–92.29)	*p <* 0.001	*p <* 0.001	*p <* 0.001	*p <* 0.001	0.641	−0.985
Pressure pain sensitivity index fingers (N) (0–100)	37.64 ± 150.02 (33.49–41.78)	380.06 ± 14.51 (33.91–42.20)	78.78 ± 18.89 (74.64–82.93)	*p <* 0.001	*p*>0.05	*p <* 0.001	*p <* 0.001	0.589	−0.028
Vibration thresholds index fingers (vibration units)	3.52 ± 0.82 (3.33–3.70)	3.21 ± 0.81 (30.02–3.39)	1.61 ± 0.50 (1.42–1.79)	*p <* 0.001	*p <* 0.05	*p <* 0.001	*p <* 0.001	0.573	0.368
Vibration thresholds toes (vibration units)	4.51 ± 0.81 (4.29–4.71)	4.14 ± 10.01 (3.94–4.35)	2.18 ± 0.56 (1.97–2.38)	*p <* 0.001	*p <* 0.05	*p <* 0.001	*p <* 0.001	0.615	0.404
Berg scale (0–56)	28.12 ± 4.84 (26.77–29.46)	33.15 ± 7.59 (31.80–34.49)	55.57 ± 1.63 (54.22–56.91)	*p <* 0.001	*p <* 0.005	*p <* 0.001	*p <* 0.001	0.838	−0.792
Six-minute walking test (m)	363.8 ± 61.48 (344.3–383.2)	401.3 ± 93.70 (381.8–420.7)	611.7 ± 70.17 (592.2–631.1)	*p <* 0.001	*p <* 0.005	*p <* 0.001	*p <* 0.001	0.675	−0.473
Timed Up and Go Test (s)	17.58 ± 4.83 (16.70–18.46)	13.18 ± 3.20 (12.30–140.06)	7.70 ± 1.56 (6.82–8.58)	*p <* 0.001	*p <* 0.005	*p <* 0.005	*p <* 0.005	0.580	10.074
Isometric back muscle strength (kiloponds)	29.25 ± 9.81 (23.59–34.90)	420.08 ± 25.70 (36.43–47.73)	120.4 ± 26.81 (114.7–1260.0)	*p <* 0.001	*p <* 0.05	*p <* 0.001	*p <* 0.001	0.770	−0.660
Borg scale (0–10)	6.63 ± 1.48 (6.27–6.99)	40.02 ± 1.67 (3.65–4.38)	10.09 ± 10.06 (0.73–1.46)	*p <* 0.001	*p <* 0.001	*p <* 0.001	*p <* 0.001	0.719	1.654
Mean sway velocity (cm/s)	0.019 ± 0.009 (0.017–0.020)	0.011 ± 0.005 (0.010–0.013)	0.007 ± 0.001 (0.006–0.009)	*p <* 0.001	*p <* 0.001	*p <* 0.001	*p <* 0.001	0.363	1.143
Mediolateral body sway (cm)	0.013 ± 0.009 (0.011–0.014)	0.007 ± 0.005 (0.005–0.008)	0.003 ± 0.001 (0.001–0.005)	*p <* 0.001	*p <* 0.001	*p <* 0.01	*p <* 0.01	0.261	0.714
Anteroposterior body sway (cm)	0.015 ± 0.008 (0.014–0.017)	0.011 ± 0.006 (0.009–0.012)	0.008 ± 0.002 (0.006–0.009)	*p <* 0.001	*p <* 0.001	*p <* 0.05	*p <* 0.05	0.209	0.571
Gait velocity (cm/s)	2.78 ± 0.99 (2.50–30.05)	2.81 ± 10.09 (2.53–30.09)	3.67 ± 1.14 (3.40–3.95)	*p <* 0.001	*p <* 0.05	*p <* 0.02	*p <* 0.02	0.130	−0.038
Stride length (cm)	0.93 ± 0.33 (0.84–10.02)	0.97 ± 0.41 (0.88–10.06)	1.22 ± 0.34 (1.13–1.32)	*p <* 0.001	*p <* 0.05	*p <* 0.02	*p <* 0.02	0.115	−0.108
Percentage of time in the stance phase (%)	67.17 ± 10.42 (65.31–690.04)	68.44 ± 5.49 (66.58–70.31)	65.38 ± 4.62 (63.52–67.24)	*p <* 0.001	*p <* 0.05	*p <* 0.02	*p <* 0.02	0.029	−0.152
Percentage of time in the swing phase (%)	31.66 ± 5.87 (30.29–330.04)	31.69 ± 5.62 (30.31–330.07)	34.58 ± 4.64 (33.21–35.96)	*p <* 0.001	*p <* 0.05	*p <* 0.02	*p <* 0.02	0.062	−0.005

FM: fibromyalgia; CLBP: chronic low back pain; N: newtons; cm: centimeters; s: seconds.

## Data Availability

The data presented in this study are available on request from the corresponding author.

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
