# Peer review of "An Observational Study Comparing Fibromyalgia and Chronic Low Back Pain in Somatosensory Sensitivity, Motor Function and Balance"

_healthcare, 2021, doi:10.3390/healthcare9111533_

Round 1

Reviewer 1 Report

see attached file

Author Response

This study aimed at evaluating pain, pain sensitivity and motor function in fibromyalgia patients (FM) vs chronic low back patients (CLBP) as compared to normal subjects. Somatosensory sensitivity and motor function were worse in both patients’ groups vs controls. In FM there were greater pain impact and somatosensory and motor deficits than in CLBP.

The study is very interesting and of great clinical impact as it addresses two major chronic pain conditions and the possible underlying factors of the transition of one (CLBP) into the other (FM). The experimental design is well conceived, the paper very well written and easy to follow.

Thanks.

My suggestions are minor, for further improvement:

-Pain sensitivity may vary depending on the phase of the menstrual cycle in women. Were all women of the study in post-menopause or were there any patients/subjects still in their fertile phase of life ? If so, were sensory evaluations performed always in the same relative phase of the cycle ?

Thanks for the observation. Although most patients were post-menopausal, this fact was not considered in our study, so it has been added as a limitation (page 13). This factor will be considered for future studies.

-Although the main focus of the study is on pain and motor function, also other different symptoms have emerged from the analysis, among which depression. Since depression is a crucial aspect in chronic pain, it would be of interest if the authors could expand the discussion on this aspect, particularly with respect to  the relationship between depression and the opioid system [see, for instance Peripheral blood mononuclear cell beta-endorphin concentration is decreased in chronic fatigue syndrome and fibromyalgia but not in depression: preliminary report. Clin J Pain. 2002;18(4):270-3)]

Thanks. This point has been added to the discussion (page 13), along with the inclusion of bibliographic references.

Reviewer 2 Report

The manuscript entitled “Fibromyalgia and chronic low back pain. A comparison be-2 tween two chronic pain entities” aims to compare pain impact, somatosensory sensitivity, motor functionality and balance among patients with fibromyalgia, patients with chronic low back pain and pain-free controls. I have revised the manuscript and provide some comments and suggestions which I hope the Authors will revise. The following comments are offered to help strengthen the manuscript:

Title:

The title “Fibromyalgia and chronic low back pain. A comparison between two chronic pain entities” does not provide information about the design of the study.

Abstract section:

The methods are not described in the abstract.

Methods section:

The design of the study is not stated.

Number of participants included in the study should be reserved to the results section.

Information provided in “Table 1. Sociodemographic data from the three groups of participants (fibromyalgia, chronic low back pain and pain-free controls). BMI: body mass index.” should be provided in the results section.

The methods should describe the methodology used in the study and the results the data obtained. Please revise.

Abbreviations in tables should be provided as footnote of the table.

Units of BMI in table 1 are not reported.

Why was executive function evaluated with Berg scale, six minute walking test, timed up and go test, isometric back muscle strength and borg scale? Please explain the relation of these scales with executive function. This term “executive function” is used as keyword and as title to group these tests; however it is not explained in the manuscript the importance or connection between the tests to determine executive function.

Please explain P value of 0.06 in gender (table 1). All participants were women.

One of the EQUATOR guidelines should be used to carry out the research.

No references for the assessment of dynamic balance are provided.

Results section:

Tables showing the mean and standard deviation as well as p-values should be provided.

Conclusion section

The conclusion does not clearly answer to the objective.  

Author Response

The manuscript entitled “Fibromyalgia and chronic low back pain. A comparison be-2 tween two chronic pain entities” aims to compare pain impact, somatosensory sensitivity, motor functionality and balance among patients with fibromyalgia, patients with chronic low back pain and pain-free controls. I have revised the manuscript and provide some comments and suggestions which I hope the Authors will revise. The following comments are offered to help strengthen the manuscript:

Title:

The title “Fibromyalgia and chronic low back pain. A comparison between two chronic pain entities” does not provide information about the design of the study.

The title has been changed, providing information on the study design.

Abstract section:

The methods are not described in the abstract.

The methods have been described in the abstract.

Methods section:

The design of the study is not stated.

Study design added.

Number of participants included in the study should be reserved to the results section. Information provided in “Table 1. Sociodemographic data from the three groups of participants (fibromyalgia, chronic low back pain and pain-free controls). BMI: body mass index.” should be provided in the results section. The methods should describe the methodology used in the study and the results the data obtained. Please revise.

The number of participants and Table 1 have been moved to the results section.

Abbreviations in tables should be provided as footnote of the table.

Units of BMI in table 1 are not reported.

Table footnotes have been provided and BMI units have been reported in Table 1.

Why was executive function evaluated with Berg scale, six minute walking test, timed up and go test, isometric back muscle strength and borg scale? Please explain the relation of these scales with executive function. This term “executive function” is used as keyword and as title to group these tests; however it is not explained in the manuscript the importance or connection between the tests to determine executive function.

Executive function has been removed as a keyword and it has been replaced by motor function in the main text.

Please explain P value of 0.06 in gender (table 1). All participants were women.

This was a typo. Not all participants were females (54 women in FM, 45 in CLBP, and 45 in control group).

One of the EQUATOR guidelines should be used to carry out the research.

The STROBE guide has been used to assist in conducting the research.

No references for the assessment of dynamic balance are provided.

References for dynamic balance have been added.

Results section:

Tables showing the mean and standard deviation as well as p-values should be provided.

Table 2 has been added showing the means, standard deviation and p-values of all variables.

Conclusion section

The conclusion does not clearly answer to the objective.

We have tried to define our objectives more clearly and we have modified some aspects of the discussion, trying to make this link clearer.

Reviewer 3 Report

Thank you for the opportunity to review this paper. In essence, this paper compares patients with chronic low back pain, fibromyalgia and pain-free controls in terms of several sensory, motor and functional outcomes. I found the topic interesting, and the results seem to be novel.

The main limitation, at least in my view, is not inherent to the study, but to the potential heterogeneity of the pain conditions, especially LBP. The LBP could arise for several different reasons, and the underlying reasons could be differently affecting the scores of the tests you used. It is good that you did some efforts to narrow down the sample (e.g., excluding patients that underwent surgery). But still, the potential heterogeneity between patients is high. At the minimum, the descriptive statistics for all outcome parameters should be included (not only Means and SD, please also include min-max ranges). Moreover, this limitation should be heavily stressed in the discussion.

Other comments:

Abstract.

Please include at least the group sample sizes and age range.

I would also suggest you incorporate at least the key results in numerical form into the abstract. If there is too little space due to journal limitations, try to include least the effect sizes (numerically or descriptively)

Introduction

Line 35 – Delete the “unknown” … the word idiopathic should be understood by the readership of this journal

Line 48-49 – “anticipatory postural adjustments and increased oscillations of the center of gravity” This is not the best writing… APAs are one of the stability mechanisms, while increased oscillations are one of the outcome variables used to quantify… please write both at the level of control mechanisms (i.e. saying that both APA and steady-state balance are impaired).

Line 51-53 – Again, similar comment. You list specific parameters (e.g. stride length) and groups of variables (biomechanical parameters). Either outline all specific parameters that were found to be impaired, or generalize everything.

Line 55-56 – It is not clear how the results of this study will help in diagnostic process. Could you perhaps try to elaborate further?

Line 61. A reference should be included about this fact

Methods

You did a good job with the inclusion/exclusion criteria. But these do not tell everything about the final sample characteristics. The description of the clinical characteristics of the participants is lacking (eg: average pain-level, disability, duration of pain, …), and these clinical aspects are not taken into account in the statistical analyses.

Line 76. Please confirm and describe (if it is true) that this means that the patients could only be included in the study if had sought medical help for their problems before.

Lines 106-114. Were the assessments performed in the same order for all participants or was the order randomized?

Line 129 and 139. Were there any breaks and how long between the trials?

Line 148. Describe in additional sentence the set-up. Was the set-up in 10x5 m square, as in SFT test battery, or somehow different?

Line 151. Number of repetitions and breaks?

The sampling rate of the camera to capture body sway is very low and should be heavily stressed as a limitation of the study.

Line 196. Which version of SPSS?

Line 202. Add the thresholds for the magnitude of correlations (e.g., 0.00-0.1; trivial, 0.1-0.4, small, 0.4-0.7, moderate, and so on…), and an appropriate reference

Effect sizes should be added for ANOVA (eta-squared), and, if possible, Cohen’s d as well at least for the most important pos hoc comparisons.  

Results

The most urgent issue is the lack of some data. Descriptive statistics is only provided for some parameters (figures 1 and 2). Please add the complete data, either by inserting additional tables, figures and/or text. I recommend you include everything in a Table (descriptive statistics as well as differences test).

I also wonder about the isometric strength assessment. As I understand, you measured and reported force? I think it would be appropriate to normalize it to the body mass before comparing the groups.

Line 317-319. Please delete this extra text

A final note on results – you say in the methods that you will analyze the relationship between kinematic parameters and clinical symptoms in chronic pain (FM and CLBP) – where are these results?

Discussion

Generally, the discussion is well written.

However, the rationale of the study was explained with one sentence in the introduction (“Establishing an accurate diagnostic profile can help clinicians interpret these complex clinical patterns.”). Fair enough, but please explain in the discussion how your results help.

Author Response

Thank you for the opportunity to review this paper. In essence, this paper compares patients with chronic low back pain, fibromyalgia and pain-free controls in terms of several sensory, motor and functional outcomes. I found the topic interesting, and the results seem to be novel.

Thanks.

The main limitation, at least in my view, is not inherent to the study, but to the potential heterogeneity of the pain conditions, especially LBP. The LBP could arise for several different reasons, and the underlying reasons could be differently affecting the scores of the tests you used. It is good that you did some efforts to narrow down the sample (e.g., excluding patients that underwent surgery). But still, the potential heterogeneity between patients is high. At the minimum, the descriptive statistics for all outcome parameters should be included (not only Means and SD, please also include min-max ranges). Moreover, this limitation should be heavily stressed in the discussion.

Table 2 has been added and this point has also been included in the discussion, within the limitations of this study.

Other comments:

Abstract.

Please include at least the group sample sizes and age range.

Sample size and age range have been included.

I would also suggest you incorporate at least the key results in numerical form into the abstract. If there is too little space due to journal limitations, try to include least the effect sizes (numerically or descriptively)

Key results have been incorporated into the abstract.

Introduction

Line 35 – Delete the “unknown” … the word idiopathic should be understood by the readership of this journal.

The word “unknown” has been deleted.

Line 48-49 – “anticipatory postural adjustments and increased oscillations of the center of gravity” This is not the best writing… APAs are one of the stability mechanisms, while increased oscillations are one of the outcome variables used to quantify… please write both at the level of control mechanisms (i.e. saying that both APA and steady-state balance are impaired).

We have rewritten the introduction a bit.

Line 51-53 – Again, similar comment. You list specific parameters (e.g. stride length) and groups of variables (biomechanical parameters). Either outline all specific parameters that were found to be impaired, or generalize everything.

Everything has been generalized as biomechanical parameters.

Line 55-56 – It is not clear how the results of this study will help in diagnostic process. Could you perhaps try to elaborate further?

We have tried to elaborate this point at the end of the Conclusions.

Line 61. A reference should be included about this fact.

The reference has been added.

Methods

You did a good job with the inclusion/exclusion criteria. But these do not tell everything about the final sample characteristics. The description of the clinical characteristics of the participants is lacking (eg: average pain-level, disability, duration of pain, …), and these clinical aspects are not taken into account in the statistical analyses.

Clinical characteristics of the participants with chronic pain have been added as inclusion criteria.

Line 76. Please confirm and describe (if it is true) that this means that the patients could only be included in the study if had sought medical help for their problems before.

It has been specified that patients with chronic pain should have a clinical diagnosis.

Lines 106-114. Were the assessments performed in the same order for all participants or was the order randomized?

It has been specified that the evaluations were carried out in the same order for all participants.

Line 129 and 139. Were there any breaks and how long between the trials?

It has been specified that there were no breaks between trials.

Line 148. Describe in additional sentence the set-up. Was the set-up in 10x5 m square, as in SFT test battery, or somehow different?

It was added that the set-up was a 30-meter corridor before the entrance to the room with place cones at either end of the 30-metre stretch as turning points.

Line 151. Number of repetitions and breaks?

It was added that this task was performed only once.

The sampling rate of the camera to capture body sway is very low and should be heavily stressed as a limitation of the study.

This was a typo and has been corrected. The camera recorded at 210 frames per second.

Line 196. Which version of SPSS?

It was added the version.

Line 202. Add the thresholds for the magnitude of correlations (e.g., 0.00-0.1; trivial, 0.1-0.4, small, 0.4-0.7, moderate, and so on…), and an appropriate reference.

This was a typo. Those correlations were not made.

Effect sizes should be added for ANOVA (eta-squared), and, if possible, Cohen’s d as well at least for the most important pos hoc comparisons.

This data has been added to table 2.

Results

The most urgent issue is the lack of some data. Descriptive statistics is only provided for some parameters (figures 1 and 2). Please add the complete data, either by inserting additional tables, figures and/or text. I recommend you include everything in a Table (descriptive statistics as well as differences test).

It has been added table 2.

I also wonder about the isometric strength assessment. As I understand, you measured and reported force? I think it would be appropriate to normalize it to the body mass before comparing the groups.

It has been added that the force data were presented normalized for body size and the reference that was used has been provided.

Line 317-319. Please delete this extra text

Extra text has been removed.

A final note on results – you say in the methods that you will analyze the relationship between kinematic parameters and clinical symptoms in chronic pain (FM and CLBP) – where are these results?

This was a typo. Those correlations were not made.

Discussion

Generally, the discussion is well written.

Thanks.

However, the rationale of the study was explained with one sentence in the introduction (“Establishing an accurate diagnostic profile can help clinicians interpret these complex clinical patterns.”). Fair enough, but please explain in the discussion how your results help.

We have tried to clarify this issue in the Introduction and in the Discussion.

Reviewer 4 Report

The topic of the paper is of interest, and the work / the efforts the authors have invested in are well documented. The authors provide a good overview of fibromyalgia and chronic low back pain. However, there are some weaknesses in the presentation. 

1. Check the correct reference format according to the author guidelines of this journal.
2. It is better not to use abbreviations for terms that are not mentioned again.
3. Check for format consistency in each cell of the table, and check for missing abbreviations under the table.
4. According to Table 1. on page 3, there was no difference between groups in age. It was said that the age of the participants was between 30 and 65 years old. I wonder if meaningful results will be obtained if we compare and analyze by age group.
5. According to Table 1. on page 3, there was no difference between groups in terms of gender. There were 60 subjects in each group, and among them, 54 women in the FB group, 45 in the CLBP group, and 45 in the Pain-free control group. There were 54 participants in the FB group, but it seems to be 60 women. Also, there seems to be differences in the senses such as sensory thresholds and somatosensory sensitivity depending on gender.

Author Response

Comments and Suggestions for Authors

The topic of the paper is of interest, and the work / the efforts the authors have invested in are well documented. The authors provide a good overview of fibromyalgia and chronic low back pain.

Thanks.

However, there are some weaknesses in the presentation. 

  1. Check the correct reference format according to the author guidelines of this journal.

The reference format has been verified.
2. It is better not to use abbreviations for terms that are not mentioned again.

Abbreviations for terms that are not mentioned again have been removed.
3. Check for format consistency in each cell of the table, and check for missing abbreviations under the table.

The consistency of the table has been checked and the abbreviations below the table have been verified.
4. According to Table 1. on page 3, there was no difference between groups in age. It was said that the age of the participants was between 30 and 65 years old. I wonder if meaningful results will be obtained if we compare and analyze by age group.

Due to the small sample of the study, we have considered that a comparison by age (at least two levels, middle-aged and older adults) and group (3 levels) would lack sufficient statistical power.

  1. According to Table 1. on page 3, there was no difference between groups in terms of gender. There were 60 subjects in each group, and among them, 54 women in the FB group, 45 in the CLBP group, and 45 in the Pain-free control group. There were 54 participants in the FB group, but it seems to be 60 women.

Table 1 clarifies the number of men and women in each of the groups.

Also, there seems to be differences in the senses such as sensory thresholds and somatosensory sensitivity depending on gender.

Due to the small number of males in the study, we have considered that a comparison by gender and group (3 levels) would lack sufficient statistical power.

Reviewer 5 Report

Dear author,

The article ‘’Fibromyalgia and chronic low back pain. A comparison between two chronic pain entities’’ is well written, nevertheless i invite you to increase the paper. 
Preferably the length of a published Research articles should be of 4000-6000 words with 75 or more references excluding figures, structures, photographs, schemes, tables, etc. 
I suggest you to increase the introduction focusing on possible differential diagnosis with other pathology with similar somatosensory and motor symptoms as stiffness and tenderness in muscles, tendons and joints, as well as sleep disturbances, fatigue, etc. in addition shows the differences trough the gold standard method for CLBP diagnosis, FM diagnosis and other pathology (TMD, Orofacial pain, hypersensitivity, oral medicine etc.)

These articles could be useful:

  • Temporomandibular disc displacement with reduction treated with anterior repositioning splint: A 2-year clinical and magnetic resonance imaging (MRI) follow-up PMID: 32064850

  • Unilateral superior condylar neck fracture with dislocation in a child treated with an acrylic splint in the upper arch for functional repositioning of the mandible   PMID: 27398739

  • A Comprehensive Review of Over the Counter Treatment for Chronic Low Back Pain PMID: 33150555

  • Effectiveness of osteopathic interventions in chronic non-specific low back pain: A systematic review and meta-analysis.PMID: 33197571 

  • Polyphenols as potential agents in the management of temporomandibular disorders

    DOI: 10.3390/APP10155305

  • Mandibular coronoid process hypertrophy: Diagnosis and 20-year follow-up with CBCT, MRI and EMG evaluations DOI: 10.3390/app11104504

  • Oral-facial-digital syndrome (OFD): 31-year follow-up management and monitoring PMID: 29460530

  • Systemic and topical photodynamic therapy (PDT) on oral mucosa lesions: An overview

  • A new combined protocol to treat the dentin hypersensitivity associated with non.carious cervical lesions: A randomized controlled trial

In addition add an excursus of the most frequent therapies (TENS; physical therapy ; flat spint; radio-frequency; etc)

Finally the readers would be facilitated in reading by List of abbreviations.

Best Regards 

Author Response

Dear author,

The article ‘’Fibromyalgia and chronic low back pain. A comparison between two chronic pain entities’’ is well written, nevertheless i invite you to increase the paper. 

Thanks.
Preferably the length of a published Research articles should be of 4000-6000 words with 75 or more references excluding figures, structures, photographs, schemes, tables, etc.  I suggest you to increase the introduction focusing on possible differential diagnosis with other pathology with similar somatosensory and motor symptoms as stiffness and tenderness in muscles, tendons and joints, as well as sleep disturbances, fatigue, etc. in addition shows the differences trough the gold standard method for CLBP diagnosis, FM diagnosis and other pathology (TMD, Orofacial pain, hypersensitivity, oral medicine etc.)

We sincerely appreciate the reviewer's comments, but we believe that extending the Discussion to other chronic pain states such as temporomandibular disorders, orofacial pain, etc., would be outside the scope of our study. As we have indicated in the Introduction, the objective of the study was to make a comparison between patients with CLBP and FM, since they are two chronic pain pathologies with similar clinical characteristics and motor dysfunctions that were not previously compared.

These articles could be useful:

  • Temporomandibular disc displacement with reduction treated with anterior repositioning splint: A 2-year clinical and magnetic resonance imaging (MRI) follow-up PMID: 32064850
  • Unilateral superior condylar neck fracture with dislocation in a child treated with an acrylic splint in the upper arch for functional repositioning of the mandible   PMID: 27398739
  • A Comprehensive Review of Over the Counter Treatment for Chronic Low Back Pain PMID: 33150555
  • Effectiveness of osteopathic interventions in chronic non-specific low back pain: A systematic review and meta-analysis.PMID: 33197571 
  • Polyphenols as potential agents in the management of temporomandibular disorders. DOI: 10.3390/APP10155305
  • Mandibular coronoid process hypertrophy: Diagnosis and 20-year follow-up with CBCT, MRI and EMG evaluations DOI: 10.3390/app11104504
  • Oral-facial-digital syndrome (OFD): 31-year follow-up management and monitoring PMID: 29460530
  • Systemic and topical photodynamic therapy (PDT) on oral mucosa lesions: An overview
  • A new combined protocol to treat the dentin hypersensitivity associated with non.carious cervical lesions: A randomized controlled trial

In addition add an excursus of the most frequent therapies (TENS; physical therapy ; flat spint; radio-frequency; etc)

Once again, we welcome the reviewer's comments, but we believe that extending the Discussion to common therapies used in CLBP and FM would not add meaningful information to the main topic of the article: are FM patients more impaired than CLBP patients with respect to somatosensory sensitivity and gait and balance parameters?

Finally the readers would be facilitated in reading by List of abbreviations.

Abbreviations have been removed from the text, except for chronic low back pain (CLBP) and FM (fibromyalgia), which were included in the text once mentioned.

Round 2

Reviewer 2 Report

The manuscript entitled “Fibromyalgia and chronic low back pain. A comparison between two chronic pain entities” aims to compare pain impact, somatosensory sensitivity, motor functionality and balance among patients with fibromyalgia, patients with chronic low back pain and pain-free controls. I have revised the new version of this  manuscript and provide some comments and suggestions which I hope the Authors will revise. The following comments are offered to help strengthen the manuscript:

Methods section:

Why was Borg scale included as an instrument used to evaluate motor function?

Results section:

Please discuss P value of 0.06 in age and gender (table 1). P value is near to the 0.05 this could be a limitation and interfere in the results.

P-values in tables 1 and 2 are referred to the comparison of the three groups? Where are the p-values between each pair of groups of patients? It would be useful to show these results in tables.

Author Response

The manuscript entitled “Fibromyalgia and chronic low back pain. A comparison between two chronic pain entities” aims to compare pain impact, somatosensory sensitivity, motor functionality and balance among patients with fibromyalgia, patients with chronic low back pain and pain-free controls. I have revised the new version of this  manuscript and provide some comments and suggestions which I hope the Authors will revise. The following comments are offered to help strengthen the manuscript:

Methods section:

Why was Borg scale included as an instrument used to evaluate motor function?

The Borg scale was used to assess perceived exertion after each physical activity task. This widely used scale provides reliable estimates exertion intensity during exercise and is applicable for both healthy persons and individuals with chronic diseases.

Results section:

Please discuss P value of 0.06 in age and gender (table 1). P value is near to the 0.05 this could be a limitation and interfere in the results.

We have included two sentences at the end of the Discussion (lines 427-432), adding this issue as a further limitation of the study.

P-values in tables 1 and 2 are referred to the comparison of the three groups? Where are the p-values between each pair of groups of patients? It would be useful to show these results in tables.

P-values between each pair of patient groups have been added to Tables 1 and 2.

Reviewer 3 Report

Thank you for addressing my comments. No further questions from me. 

Author Response

In the final version of the manuscript, changes have been made in the specified sections, with the aim of improving them. Thanks.